# Monitoring the emotional facial reactions of individuals with antisocial personality disorder during the retrieval of self-defining memories

Audrey Lavallee[1,2,3], Thierry. H. Pham[2,4], Marie-Charlotte Gandolphe[1], Xavier Saloppé[1,4,5], Laurent Ott[1], Jean-Louis Nandrino [1]*

1 SCALab, UMR CNRS 9193, University of Lille, Villeneuve d'Ascq Cedex, France, 2 Department of Legal Psychology, University of Mons, Mons, Belgium, 3 Groupement des Hôpitaux de l'Institut Catholique de Lille, Lille, France, 4 Research Center in Social Defense, Tournai, Belgium, 5 Psychiatric Hospital, Saint-Amand-les-Eaux, France

* jean-louis.nandrino@univ-lille.fr

**Data Availability Statement:** All relevant data are within the paper and its Supporting Information files.

## Abstract

While a deficit in the recognition of facial expression has been demonstrated in persons with antisocial personality disorder (ASPD), few studies have investigated how individuals with ASPD produce their own emotional facial expressions. This study examines the production of facial emotional expressions of male inpatients with ASPD in a forensic hospital compared with a control group as they retrieve autobiographical memories. This design constitutes a specific ecological experimental approach fostering the evocation of personal feelings. Two indicators characterizing the activation of facial expression were used: activation of emotional action units and emotional dominance. The results showed that individuals with ASPD 1) activated angrier facial expressions than control participants for both indicators, 2) displayed a higher dominance of angry facial expressions during the retrieval of positive self-defining memories than control participants and 3) recalled significant memories that were less associated with neutral facial states than the control sample, regardless of the valence of their memories. These findings highlight the core role of anger in ASPD and the possible development of pathological anger, which would distinguish trajectories toward anxious or mood disorders and trajectories characterized by external disorders.

## Introduction

Facial expressions (FEs) are part of body language and can spontaneously convey internal experiences or be intentionally adjusted to allow social communication, interaction and regulation [1–4]. While research has mainly focused on the ability to recognize emotional facial expressions and has identified difficulties in FE recognition in various mental and personality disorders [5–8], fewer studies have investigated the ability to produce emotional FEs in personality disorders, especially in individuals with antisocial personality disorders (ASPD) [9, 10]. However, ASPD is associated with emotional disturbance, and subjects with ASPD are

**Funding:** This work was funded by the French National Research Agency (ANR-11-EQPX-0023) and was also supported by European funding through the program FEDER SCV-IrDIVE.

**Competing interests:** The authors have declared that no competing interests exist.

generally described as self-centered, impulsive and callous. They find it difficult to conform to laws or norms, lack empathy and concern for others and encounter difficulties in having a stable relationship [11].

Fanti et al. [9] examined emotional facial responses in the general population with varying levels of impulsive aggression, which is a trait of ASPD, while they watched a violent film or comedy show. They found that participants with a high level of impulsive aggression globally expressed more anger than individuals with a low level of impulsive aggression and more anger regardless of the type of film (comedy or violent) [9]. Consistent with studies on hostile attribution in ASPD subjects [12, 13], the authors reported evidence of the specific status of anger in persons with a high level of impulsive aggression who might misinterpret their environment as generally threatening and for whom anger is like a personality trait. Even though anger seems to play a core role in individuals with ASPD from a clinical point of view [14], the scientific literature is limited and unclear about how individuals with ASPD experience anger. Contrary to the findings of Fanti et al. [9], Künecke et al. [10] found that antisocial offenders with different levels of psychopathy assessed in German forensic-psychiatric hospitals and correctional facilities did not differ from control participants on their corrugator muscle response to the observation of the FEs of others. Moreover, Lobbestael et al. [15] asked ASPD participants to recall a conflictual event and to describe how they felt and what they wanted to do during it. Although they expected that individuals with ASPD would deny their anger, the participants did not express any abnormal anger. Nevertheless, the authors argued that it was difficult to measure anger objectively and considered that individuals with ASPD should experience more anger than what they found in their study. In that same line, the study of emotional expression in patients with ASPD is particularly important because it allows us to examine the ability to communicate one's own emotional states and to explain possible difficulties in social interactions, especially in inducing fearful or aggressive responses. In these disorders, it is necessary to better articulate the person's subjective emotional experience and what they are able to communicate about it.

Most studies of the emotional experiences of ASPD subjects have used experimental designs, and their emotional responses were not studied from personal material but from image bases or external media. To our knowledge, no study has examined the spontaneous and personal emotional experiences arising from their own life history. The present study thus examined the emotional responses of participants with ASPD during the retrieval of self-defining memories (SDM) to mirror their own experiences as closely as possible.

SDMs are particular autobiographical memories that are vivid, emotionally intense, frequently recalled and concern long-lasting or unresolved issues [16]. They are particularly associated with the construction of identity and help to maintain self-coherence, particularly during times of difficult transitions or upheaval [17]. Compared with autobiographical memories, SDMs are subjectively more important, emotionally more activating and are associated more with events playing a central role in the construction of identity [18]. Thus, SDMs are appropriate for exploring the expressivity of spontaneous and personal emotions and provide an ecological paradigm to analyze the emotions felt by persons with ASPD.

As oral tasks rule out the use of electromyography because movements due to nonemotional contractions or nonverbal communication cause interference in data collection [19–21], we assessed facial responses using facial coding software: FaceReader [22]. Unlike electromyography, which may be obtrusive because of the electrodes applied to the face, FaceReader is a noninvasive technology that requires the use of a video camera. It accounts for the activation of several emotional action units and for the joint activation of several facial muscles corresponding to a given emotional facial expression. In the past decade, it has been used widely in a variety of contexts, such as psychology [5, 23, 24], social psychology [25] and advertising research [26]. It is a facial analysis program based on the Facial Action Coding System (FACS,

[27, 28]). FACS is based on the visually discernible movement of a set of facial muscles referred to as 44 action units and on several head and eye movements. The joint activation of action units and their intensity may thus be regarded as the expression of one of the six basic emotions (joy, anger, sadness, fear, disgust, and surprise). FaceReader thus overcomes the limitation of electromyography, which cannot clearly distinguish different types of emotions, notably because of the contraction of the corrugator muscle in several negative emotions. Thus, FaceReader makes it possible to discriminate the activation of basic emotions at least as well as expert humans and more quickly [25, 29]. Lewinski et al. [29] proved the reliability of FaceReader to categorize the activation of basic emotions, and Gandolphe et al. [30] demonstrated its effectiveness for capturing FEs during the retrieval of SDMs.

The present study, therefore, examined the emotional activation of forensic patients with ASPD during the retrieval of important personal events such as SDMs. We hypothesize that individuals with ASPD produce more anger-associated FEs than healthy participants during the retrieval of SDMs. Additionally, we expected that this overactivation of anger-associated FEs may lead to an underactivation of the other emotional or nonemotional FEs (happiness, sadness, fear and neutral).

## Methods

### Participants

Forty-two men were recruited. One group comprised 21 men with an ASPD who had committed crimes and were hospitalized in the Marronniers forensic hospital in Tournai (Belgium) (ASPD sample) (*M* age = 44.10 years; *SD* = 14.24, *M* education level = 9.48 years, *SD* = 2.62). The other comprised 21 men of the general population without mental or behavioral difficulties who were the controls (M age = 33.62, *SD* = 10.92; *M* education level = 13.90 years, *SD* = 2.48). The control sample was recruited through an advertisement distributed on social networks and from the staff of the universities of Lille in France and Mons in Belgium.

Since most ASPD participants presented conduct disorders during childhood associated with school difficulties, we were not able to match education levels between APSD and control participants. However, there was no significant age or education level effect on the action unit activation (neutral, happiness, anger, sadness and fear) between the two groups.

Concerning the ASPD sample, the inclusion criteria were as follows: a diagnosis of ASPD attested by the SCID II [31]; being a native French speaker; not having an intellectual deficiency that could interfere with the understanding of the instructions; not taking alpha-blockers and/or beta-blockers; not having neurological or medical issues that could alter memory abilities; and not being psychotic. Diagnostic scales (SCID II and PCL-R) were administered upon admission by trained professionals of the forensic hospital. Scores and diagnoses were extracted from the inpatients' medical files. A total of 65 forensic inpatients were met, 25 agreed to participate, and 21 came to the exploratory meeting. The offenses committed by the participants were as follows: sex offenses (72.72%), indecent assault (50%), robbery with (50%) or without violence (52.27%), assault (59.09%), attempted homicide (13.64%) or homicide (18.19%), torture (4.55%), abduction (9.09%), sequestration (18.18%), concealment (22.73%), drug detention (31.82%) and sale and possession of weapons (54.54%).

All participants were informed that they would not be remunerated.

### Ethical compliance and recruitment procedure

The present research was validated by the ethical committee of *Les Marronniers* forensic hospital, Tournai, Belgium (CE/DV/EA/2015) and followed the principles of the Helsinki Declaration. All participants were recruited on a voluntary basis.

After psychiatrists and psychologists of each department of the forensic hospital selected the ASPD patients on the basis of the inclusion criteria, the investigators met them to ask them if they wished to participate in the study. If they agreed, a meeting was scheduled to explain the experimental procedure to them. During this meeting, the informed consent form was read aloud by the investigator and then given to the participants. They were given time to read and think about it before signing the form. Participants in the ASPD group were informed that neither their participation nor the outcome of the study would have any impact on any legal decision affecting them or their legal status. Controls received an email containing information about the experimental procedure a few days before the first meeting. At the beginning of the meeting, the experimental paradigm was explained to them by the investigators.

After ensuring that all participants fully understood the research paradigm, they signed two consent forms, one kept by the university and the other kept by the participants themselves and in which their personal identifier code was indicated. The code allowed participants to access their personal information and request its destruction if they desired. All participants were debriefed following the study.

## Assessment of psychopathic traits

Individuals with ASPD may present varying levels of psychopathic traits ranging from totally absent to a full diagnosis of psychopathy. ASPD and psychopathy share much of their semiology, and while every person with a psychopathic personality has an ASPD, the contrary is not true [32, 33]. According to Venables et al. (2014), one of the main distinctions between these two personalities is the cold-heartedness component, which is a core characteristic of psychopathy [34]. Additionally, for descriptive purposes, the Psychopathy Checklist-Revised (PCL-R) [35, 36] score is described below. The PCL-R consists of a semistructured interview with 20 items based on a person's lifestyle functioning split into two distinct factors. Factor 1 corresponds to the interpersonal and affective components of the disorder and includes the dimensions of being selfish, callous and remorseless in the use of others and includes the dimensions of glibness and grandiosity. Factor 2 corresponds to an antisocial lifestyle and includes items relative to a chronically unstable, antisocial and socially deviant lifestyle and refers to impulsivity and sensation-seeking [36]. These two factors may be divided into four facets. Factor 1 includes facets relative to interpersonal relationships (facet 1) and affective coldness or cold-heartedness (facet 2). Factor 2 includes facets relative to impulsivity (facet 3) and to antisocial behavior (facet 4) [32].

In our ASPD sample, the mean PCL-R score was 23.89 (SD = 4.81, [12; 31]). The mean Factor 1 score was 9.35 (SD = 2.54, [5; 13]), with a mean interpersonal facet score of 3.95 (SD = 1.76, [1; 8]) and an affective facet score of 5.40 (SD = 1.82, [3; 8]). The mean Factor 2 score was 12.73 (SD = 3.88, [6; 18]), with a mean impulsivity facet score of 5.85 (SD = 1.81, [3; 9]) and an antisocial behavior facet score of 6.54 (SD = 2.85, [2; 10]).

## Retrieval of Self-Defining Memories (SDMs)

Participants were invited to retrieve five SDMs with the following instruction: "We would like you to remember five events in your life. These events must be important in defining who you are. In other words, these memories should refer to events that help you understand who you are as an individual. They should also be events that you would share with someone if you wanted that person to understand you in a basic way. The events may be positive or negative memories. The only important aspect is that they should lead to strong feelings. The memories should be events that you have thought about many times. They should also be familiar to you like a picture you have looked at a lot or a song you have learned by heart". These instructions

replicated those used in previous SDM research, especially those used by Singer and Moffit [18]. To avoid interfering with the participants' feelings, we chose not to constrain the valence or the content of their memories and left them free to choose those they wanted.

## Coding of SDM

SDMs were recorded by means of a Zoom5 microphone throughout the experiment and were transcribed and evaluated *a posteriori*. Then, their valence was objectively evaluated according to the emotional words used. If only positive emotional words were used by the participants, memories were coded as positive. They were coded as negative when the emotional words were exclusively negative [37]. If no emotional words were used, the memory was categorized as neutral. If the memory contained both positive and negative emotional vocabulary, it was coded as a mixed memory [37]. To avoid scoring bias, three independent investigators carried out the categorization, and a Cohen-Kappa coefficient was calculated to measure interrater reliability. The coefficient demonstrated an almost perfect agreement (K = 0.81) [38].

## Facial expression analyses

Memory retrieval was recorded with a charge-coupled device webcam with a resolution of $640^*480$ pixels and a 15-frame rate per second (Noldus Company) placed in front of the participant. The recording was analyzed by FaceReader 6, which analyzes a set of 20 action units described on the FACS with a 90% accuracy of recognition [39]. FaceReader 6 identifies the neutral expression and the activation of the action units relative to the six basic emotions: happiness, sadness, anger, fear, surprise and disgust. Since the lighting setup is an important criterion to provide a reliable analysis with FaceReader, a diffuse frontal light was used. For each participant and before the experimental procedure, a test phase was performed to position the camera correctly and to verify that FaceReader was able to analyze the participant's face. Because some people look angry, sad or surprised by nature, calibration was performed to correct the analyses of the software when necessary. Calibration helps FaceReader detect the neutral expression correctly and thus improves the detection of facial emotion. Calibration was performed by isolating a video fragment in which the participant was neutral. This neutral video fragment was captured during the test phase described above. Then, from this neutral video fragment, any emotional activation misanalyzed by the software was deleted. Finally, the analyses were completely reviewed according to the calibration.

In this study, the expression intensity chart of the FaceReader 6 package was used to describe any variation in the activation of the different action units during the retrieval of SDM. The values of *activation* obtained for each basic emotion were used as our first indicator. The software attributes values between 0 and 1 for each basic emotion. Value 1 corresponds to the full and exclusive activation of the combination of the action units corresponding to a basic emotion. Any other value from 0 to 1 reflects the partial activation of action units associated with basic emotion or the concurrent activation of action units related to different emotions.

Based on the values of each basic emotion, it was also possible to define the *dominant emotion*, i.e., the emotion with the significantly highest value compared with the others. To focus on the *dominant emotion*, we chose to exclude neutral expressions from this analysis. *Dominant* emotions were the second indicator. As the FEs of surprise and disgust both involve the muscles of the mouth and a raise of the brow, detection of these two emotions was hampered by the unemotional facial movements associated with language. Hence, the software did not correctly discriminate activation concerning surprise and disgust, so we excluded them from the analysis.

The data obtained with FaceReader 6 were then imported into the Observer XT software to be synchronized with the retrieval of SDM. Videos of participants were uploaded into an Observer XT file, and each SDM was computed as a behavior in the coding scheme.

## Experimental procedure

Each inpatient with ASPD was alone with two investigators in a medical examination room in the infirmary of the hospital. The room contained two desks, one where the participant and one investigator were seated opposite each other. The second investigator was at the other desk with a computer to control the software and the position of the camera in front of the participant's face. The role of the two investigators was counterbalanced across participants to give the instructions in front of the participant or to check data acquisition. During the experimental meeting with the healthy control participants, the same design as in the ASPD sample was used, except that the meetings were performed in an experimental room at the University of Lille. The investigator facing the participant was instructed not to react with his or her own emotion and to be as neutral and as supportive as possible.

## Statistical analyses

Statistical analyses were performed with R version 4.0.4. Statistical analyses were performed using Bayesian methods, on the one hand because of the experimental design and on the other because it is commonly used in autobiographical studies [40–42].

**Did the ASPD sample use the same action units as the control sample during the retrieval of SDMs?.**   We investigated whether individuals with ASPD activated the same FEs during the recall of SDMs as the general population through the expression intensity chart of FaceReader 6. The values for neutral, joy, anger, sadness, and fear FEs were obtained by averaging their activation values per SDM. To examine whether there was a group effect on the production of action units, Bayesian beta mixed-effects regression additive models (brms 2.7.0 and RStan 2.18.2 packages, [43, 44]) were performed. Beta distribution was used since values ranged from 0 to 1. Mixed-effects regressions were the most suitable, as the data were not independent, and each participant retrieved several SDMs. Bayesian analyses were the best statistical method considering sample size and data of the experimental paradigm. The independent variables of the additive models were the participants [n1; n42] (nominal random factor) and the group {ASPD, healthy control} (dichotomous fixed factor). The control group was used as a reference category for the statistical analyses. The dependent variables were the percentage of action unit activation according to the category of FEs {Neutral, Happy, Angry, Sad, Scared} (continuous numerical variable). Models were described as follows:

Model 1: Neutral, Happy, Sad, Angry, Scared ~ (Group + (1|Participants))

**Did the activation of the action units of each group vary according to the valence of the SDM retrieved?.**   SDMs were characterized according to their valence (positive, negative, neutral, mixed), and a chi-squared test was applied for intra- and intergroup comparisons. To examine whether the valence of SDMs impacted the activation of the action units, Bayesian beta mixed-effects multiplicative regression models (brms 2.7.0 and RStan 2.18.2 packages) [40, 42] were used. As in Model 1, the dependent variable was the category of FEs, and the independent variables were the Participants and the Group, but the valence of the SDMs {Positive, Negative, Neutral, Mixed} (categorical fixed factor) was added.

Model 2: Neutral, Happy, Sad, Angry, Scared ~ (Group*Valence + (1|Participant))

**Were the dominant FEs the same according to the group and/or the valence of SDM?.** Using the mean values of the activation of action units per SDM, the FE that obtained the highest value was characterized as the dominant FE of the SDM. A chi-squared test was applied to

compare the distributions within and between groups during the recall of SDMs, regardless of their valence, and a chi-squared residual test [45–47] was used to discriminate the variation with a residual value criterion greater than +/- 2 (question R 0.7.0 package, [48]). Cramer's V was used to examine the effect size. Then, to investigate whether the valence of the SDM influenced the production of FEs, Bayesian categorical logistic mixed-effects regression models (brms 2.7.0 and RStan 2.18.2 packages, [43, 44, 49]) were performed. This kind of statistical analysis is only available on Bayesian statistics with R software [49]. The independent variables were the Participants [n1; n42] (nominal random factor), Group {ASPD, Healthy Control} (dichotomous fixed factor) and Valence of SDM {Positive, Negative, Neutral, Mixed} (categorical fixed factor). The dependent variable was the Dominant FE (Dominant FE) {Happy, Angry, Sad, Scared} (multinomial categorical variable). To select the reference category in the Dominant FE variable, we chose the emotion that was mostly expressed in the control sample. For descriptive purposes about Dominant FEs in the control group, happy FEs were expressed in 47.6% of SDMs, angry FEs in 27.6%, sad FEs in 13.3% and scared FEs in 11.4%. Reference variable categories were, therefore, happy FEs (Dominant FE dependent variable), neutral SDM (valence independent variable) and control group (group independent variable).

   Model 3: Dominant_FE ~ (Group + (1|Participant))

   Model 4: Dominant_FE ~ (Group *Valence + (1|Participant))

   For the logistic model, the results indicated the difference between slopes from the reference categories and the variables of interest and could be interpreted as the odds ratios.

   For each model (beta and logistic), the results included the estimate of the probability with a 95% confidence interval (CI) (2.5%–97.5% quantiles CI). If the 95% CI does not include the zero value, then we can reject the null hypothesis and consider the estimator to be most likely different from zero and to reflect a decrease or increase in the probability as a function of the observed sign. When the credibility interval includes the value zero, the null hypothesis cannot be rejected.

## Results

Although participants were asked to retrieve 5 SDMs, 18 ASPD participants retrieved 5 SDMs, 2 recalled 4 SDMs and 1 recalled only 3 SDMs, giving a total of 101 SDMs. Furthermore, since five ASPD participants moved during the retrieval of one of their five SDMs, FaceReader was unable to analyze FE during retrieval, thus reducing the number of SDMs in this group to 96. In the control sample, all the participants recalled 5 SDMs, i.e., a total of 105 SDM.

### Did the ASPD sample use the same action units as the control sample during the retrieval of SDMs?

Model 1 shows that while the probability of not detecting the activation of emotional FEs and observing neutral action units was lower in the ASPD sample than in the control group (estimate = -0.60, 95% CI = [**-1.05; -0.18**]), it was greater regarding the activation of action units associated with angry FEs (estimate = 1.63, 95% CI = [**0.95; 2.33**]). No group effect was observed for the activation of the action units associated with happy FEs (Estimate = -0.58, 95% CI = [-1.42; 0.24]), sad FEs (Estimate = -0.47, 95% CI [-1.19; 0.26] and scared FEs (Estimate = 0.97, 95% CI = [0.09; 1.80]) (see Fig 1).

### Did the activation of the action units of each group vary according to the valence of the SDM retrieved?

In individuals with ASPD, 34.7% of the SDMs were categorized as neutral, 22.8% as positive, 21.8% as negative and 20.8% as mixed ($\chi^2$ = 2.30, $p$ = .513). In control participants,

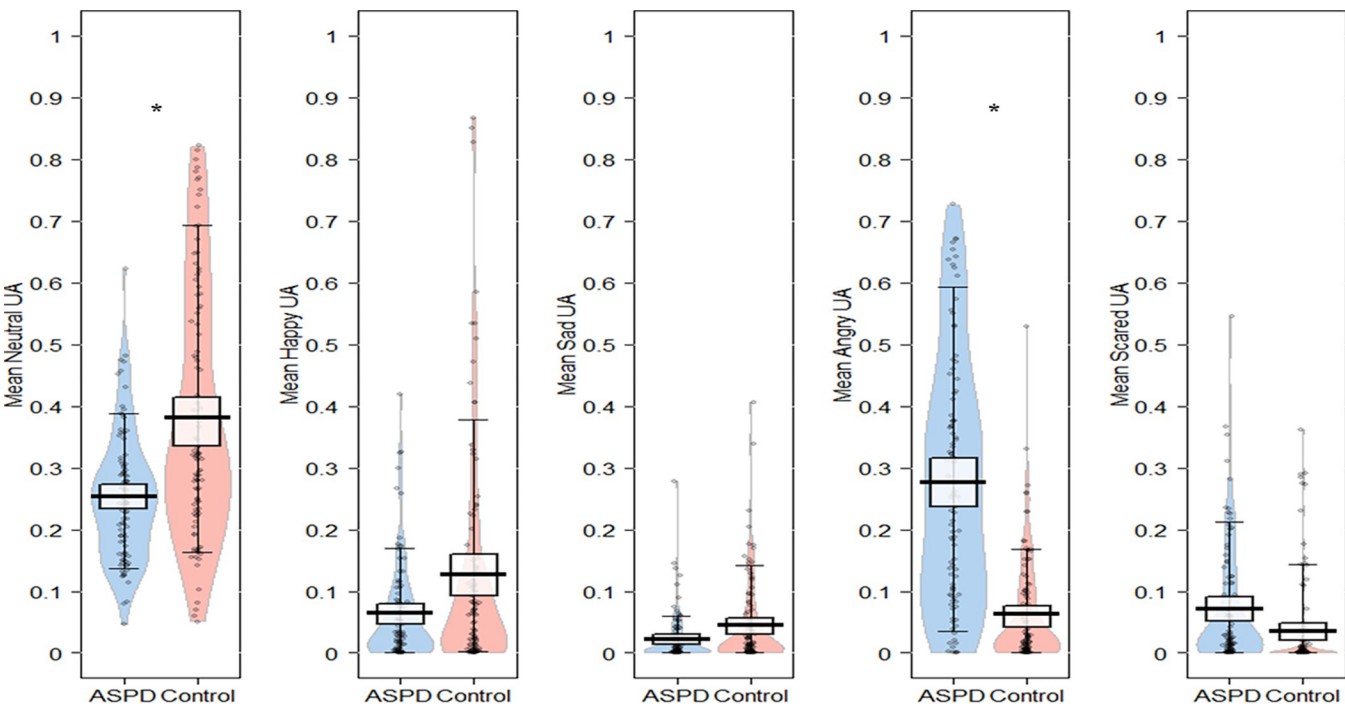

**Fig 1. Pirateplot of activation of action units per SDM in ASPD and healthy control groups.** Gray points correspond to the raw data, beans correspond to the density of distributions, thick vertical bars correspond to central tendencies of distribution and rectangle to 95% Bayesian Highest Density Intervals of distributions [50]. * corresponds to a significant result of multivariate beta regression analyses for each category of facial expressions (Model 1).

16.2% of the SDMs were categorized as neutral, 36.2% as positive, 22.9% as negative and 24.8% as mixed ($\chi^2$ = 4.23, $p$ = .238). Chi-squared tests demonstrated a difference between the two groups ($\chi^2$ = 10.46, $p$ = .015, Cramer's V = 0.23). As the valence table corresponded to a 4*2 Table (4 valences and 2 groups), it was necessary to perform a residual chi-squared test to discriminate which valence differed between our two groups [47]. However, the residual chi-squared test failed to exceed the +/- 2 criteria and in this way, does not allow us to conclude which valence of SDMs was produced more frequently by either group.

The interaction effect model (Model 2) between valence and group on the activation of action units showed that compared with the neutral SDMs, the probability of observing neutral action units was lower in the ASPD sample during the recall of mixed (Estimate = -0.47, 95% CI = [**-0.91; -0.02**]) and negative SDMs (Estimate = -0.48, 95% CI = [**-0.87; -0.07**]) than in the control group. No other interaction effect was observed. The distribution of action units according to the group and the valence of SDMs are shown in Fig 2.

### Were the dominant FEs the same according to the group and/or the valence of SDM?

Concerning the Dominant FE in control participants, happy FEs were expressed in 47.6% of SDM, angry FEs in 27.6%, sad FEs in 13.3% and scared FEs 11.4% ($\chi^2$ = 16.57, $p < .001$, Cramer's V = 0.28). In individuals with ASPD, the Dominant FE was happy in 11.5% of SDM, angry in 75%, sad in 3.1% and scared in 10.4% ($\chi^2$ = 50.93, $p < .001$, Cramer's V = 0.52). There was a difference between the distributions of the two groups ($\chi^2$ = 50.24, $p < .001$, Cramer's V = 0.50). The residual chi-square test showed differences between the ASPD and the control

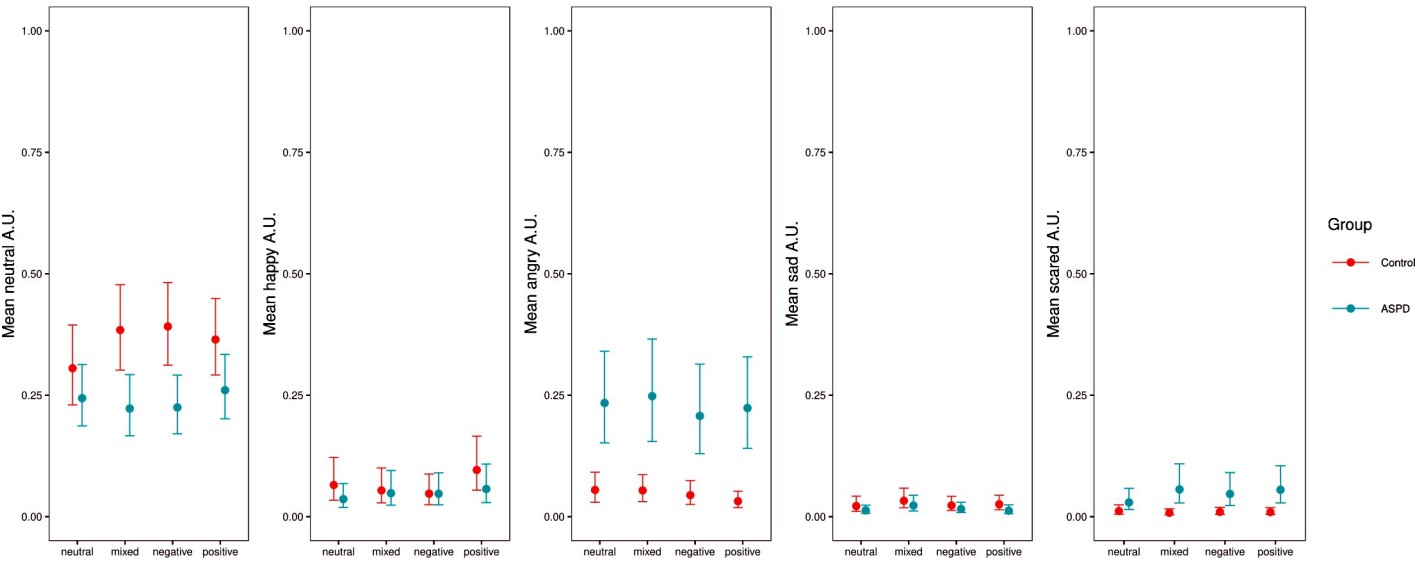

**Fig 2. Posterior distribution of the mean with a median point estimate and 95% credible interval (thinner outer lines) of Model 2 (R/conditional effects R, [49]).**

samples for the happy FE (-3.36 vs. 3.21) and for the angry FE (3.42 vs. -3.27). Distributions of the dominant emotion according to group and valence of SDM are shown in Fig 3.

Concerning the additive model of Dominant FE according to the group (Model 3), with the happy dominant FE as a reference, the probability of observing the activation of angry dominant FE was higher for the ASPD group than for the control group. No main effect was observed for the other dominant FE (sad and scared). With the happy dominant FE and the neutral SDM as references, the probability of observing the activation of the angry dominant FE during the retrieval of positive SMDs was greater for the APSD sample than for the control sample (Model 4).

The results of both models are shown in Table 1, and Model 4 is illustrated in Fig 3.

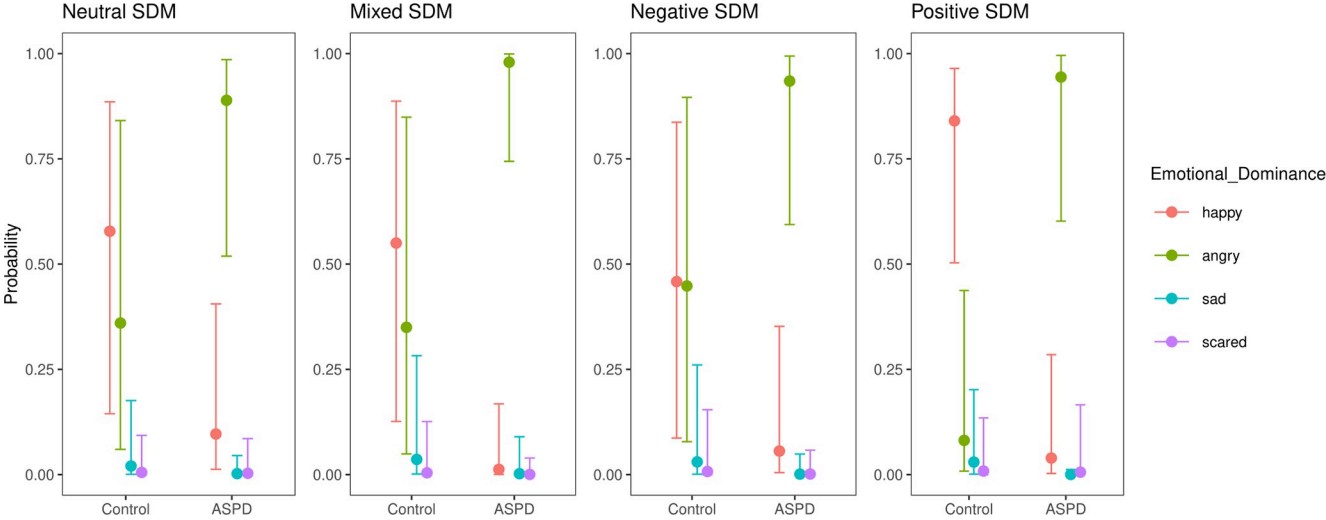

**Fig 3. Posterior distribution of the mean with a median point estimate and 95% credible interval (thinner outer lines) of Model 4 (R/conditional effects R, [49]).**

**Table 1. Results of categorical logistic regression for main effects (Model 3) and interaction effects (Model 4) models for predicting dominant facial expression in ASPD and healthy control groups and valence of SDM.**

| Group | SDM | Dominant FE | Estimate | 95% CI |
|---|---|---|---|---|
| REF = Control | REF = Neutral | REF = happy | | |
| Model 3: Dominant_FE ~ (Group + (1\|Participant)) | | | | |
| ASPD | | Angry | **3.39** | **[1.29; 5.58]** |
| | | Sad | -0.44 | [-2.96; 1.93] |
| | | Scared | 1.26 | [-1.61; 3.96] |
| Model 4: Dominant FE ~ (Group *Valence + (1\|Participant)) | | | | |
| ASPD | | | | |
| | Positive SDM | Angry | **2.87** | **[0.42; 5.37]** |
| | | Sad | -1.34 | [-4.60; 1.65] |
| | | Scared | 1.53 | [-1.21; 4.23] |
| | Negative SDM | Angry | 0.16 | [-2.12; 2.47] |
| | | Sad | -0.91 | [- 4.28; 2.28] |
| | | Scared | -0.84 | [-3.63; 1.94] |
| | Mixed SDM | Angry | 2.17 | [-0.68; 5.05] |
| | | Sad | 1.34 | [-1.81; 4.50] |
| | | Scared | 0.14 | [-3.21; 3.46] |

## Discussion

The main aim of this study was to examine the emotional experience of persons presenting with ASPD during the retrieval of personally significant events.

As expected, the results of the main effect model (Models 1 and 3) showed that the participants with ASPD activated angrier FEs more than the control participants. This activation of the angry FE cannot be explained by the emotional content of the SDMs retrieved by ASPD participants because we did not observe a significant difference in terms of valence between our two groups. The results may rather be interpreted within the framework of motivational theories characterizing anger as an approach emotion [47]. Along this line, anger is often triggered by anything that may impede the achievement of a goal. This type of emotion is opposed to fear and anxiety, which lead to avoidance rather than confrontation with the situation. If properly expressed, anger may therefore have functional benefits in removing obstacles to achieve a goal [48]. However, some people have difficulty expressing anger appropriately, leading to stabilization of the anger state and the perception of a threatening or rage-promoting environment. Recent works defend the importance of anger in personality development [50]. The ability to express anger is considered a prerequisite for the development of environmental exploration, goal achievement, preservation of personal integrity, and a sense of personal control over one's own actions [50, 51]. This specific activation of anger observed in our sample of participants with ASPD could be linked to early and repeated aversive experiences that can lead to developing pathological anger [52–55]. As Williams suggests, anger manifestations may lead to abnormal personal development and antisocial behavior when they are not systematically restrained. The development of these disorders is primarily due to two conditions related to the processing of the affective signal of anger: (a) the recurrent detection and processing of other stimuli associated with the emergence of anger or rage emotions (e.g., frustration, personal integrity violation or fear) resulting in abnormal, overly intense and/or repeated expressions of this core emotion; and (b) the confusion that occurs for the individual between his or her identity (assertiveness, autonomy, integrity, and self-control) and his or her internal emotional activation of anger [50]. This experience of anger and the difficulties in expressing it

would distinguish trajectories toward ASPD characterized by externalizing disorders and anxiety or mood disorders [14, 54, 55].

Second, we expected that persons with ASPD underexpressed their other emotions (happy, sad and scared) in comparison to healthy people. APSD participants showed fewer happy FEs and angry FEs during the retrieval of their positive SDMs than the control participants (Model 4). Although they used vocabulary associated with positive emotions in the recall of their memories, individuals with ASPD continued to express angry facial expressions (see Fig 3). This specific result reinforces the main hypothesis of the core role of anger in this disorder.

In addition, participants with ASPD showed fewer neutral FEs than the control participants (Models 1 and 2). The typical narration of a memory includes phases of contextualization and presentation of the situation before describing the actual emotional event. In control participants, it is therefore consistent to observe mainly neutral expressions, which correspond to all these descriptive and nonemotional phases of memory. In participants with ASPD, expressions of anger dominated the overall narratives (see Fig 2). Thus, they expressed fewer neutral faces than the control participants.

In light of these results, it appears that anger should be the target of psychologic interventions. When anger is maladaptive, its extreme form and notably in the setting of ASPD might lead to destructive behavior such as committing assaults or verbal attacks [51]. The latter authors posit that anger impacts the retrieval of autobiographical memory and induces the recall of more anger-related memories. Thus, anger might alter the self-construction and create a vicious circle that reinforces and maintains the symptoms of anger, overwhelms the expression and feelings of other emotions and increases the feeling of injustice.

While the present study presents the original methods and results, it has some limitations. First, the sample size reduces the generalizability of the findings and makes it impossible to make a distinction according to the level of psychopathic traits. Second, the use of SDM is specific and may also reduce the generalizability of the findings. Moreover, we did not measure the degree of congruence between FE and feelings, which should be interesting in future study projects. This comparison between facial muscle activation and subjective feeling would be important for a deeper understanding of emotions in the field of psychopathy (Kosson, personal communication). However, asking people with APSD about their subjective feelings is particularly complex, as they have a deficit in identifying and understanding them [52, 53]. Additionally, our samples included only men. As differences exist between men and women regarding the semiology of ASPD and especially their level of impulsivity [54], it would be interesting in a future study to examine whether our results can be generalized to women. Finally, we chose to focus the present study on the theory of basic emotions [28, 55] using the dominant FE as an indicator. However, other studies propose a more functional conceptualization of emotions more adapted to studying the pathological expression of emotions [56, 57]. In future studies, it would be interesting to use a more dimensional methodology to further examine the production of facial expressions during the recall of autobiographical memories. We suggest a more precise analysis of the simultaneous activation of the various action units associated with emotions during the recall of memories.

Moreover, it will be interesting to complement the measures of emotional facial reactivity made from self-defining memories with analyses of memories to emotional situations of daily life, especially because of the particular characteristics of SDMs of incarcerated patients. This type of study will help confirm whether anger hyperexpression in patients with ASPD is specific to significant personal past episodes or is generalizable to all emotional situations.

To conclude, our findings shed new light on the emotional experience of people with ASPD and notably on their emotions related to their self-construct. This study highlights the core activation of the FE associated with the emotion of anger in a forensic population with ASPD

during the retrieval of significant memories of their past. This particular activation associated with SDMs could have a role in the construction and perhaps the maintenance of ASPD and requires special attention and a specific psychological accompaniment.

## Supporting information

**S1 Data.**
(XLSX)

## Acknowledgments

The authors wish to thank the team of the EquipexIrDive, the staff of the forensic hospital the "Marronniers", Professor David Kosson for his review, and Gaëlle Husson, Fanny Degouis, Florian Sanssen and Etienne Gehenne for their valuable contribution to the project. Finally, the authors wish to thank Ray and Emily Cooke for copyediting the manuscript.

## Author Contributions

**Conceptualization:** Audrey Lavallee, Thierry. H. Pham, Xavier Saloppé, Jean-Louis Nandrino.

**Data curation:** Audrey Lavallee, Thierry. H. Pham, Xavier Saloppé.

**Formal analysis:** Audrey Lavallee, Marie-Charlotte Gandolphe, Xavier Saloppé, Laurent Ott, Jean-Louis Nandrino.

**Funding acquisition:** Thierry. H. Pham, Jean-Louis Nandrino.

**Investigation:** Marie-Charlotte Gandolphe, Jean-Louis Nandrino.

**Methodology:** Audrey Lavallee, Marie-Charlotte Gandolphe, Laurent Ott.

**Project administration:** Thierry. H. Pham, Xavier Saloppé, Jean-Louis Nandrino.

**Resources:** Thierry. H. Pham.

**Supervision:** Thierry. H. Pham, Jean-Louis Nandrino.

**Validation:** Jean-Louis Nandrino.

**Writing – original draft:** Audrey Lavallee, Marie-Charlotte Gandolphe, Xavier Saloppé, Jean-Louis Nandrino.

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
