## [Decision Letter · Decision Letter 0]

6 Sep 2021

PONE-D-21-21994Production of emotional facial expression during retrieval of self-defining memories in forensic patients with antisocial personality disorderPLOS ONE

Dear Dr. Nandrino,

Thank you for submitting your manuscript to PLOS ONE. After careful consideration, we feel that it has merit but does not fully meet PLOS ONE’s publication criteria as it currently stands. Therefore, we invite you to submit a revised version of the manuscript that addresses the points raised during the review process.

Thank you for your patience, we now received three reviews from experts in the field. They all commend the study design and the interesting approach, but they also point out a series of issues with the analysis, the discussion and the overall writing, which I agree with.

The key points across the reviews are:

- Clarifications in introduction, methods and results which are partly rather unclear

- Considering to exclude potentially high prevalence of neutral outputs from the Facereader

- Reviewing the conclusions in the discussion for validity based on the current results

- A thorough proof-reading of your manuscript.

We look forward to receiving your revised manuscript.

Kind regards,

Hedwig Eisenbarth

Academic Editor

PLOS ONE

“ This study was supported by the French National Agency for Research (grant ANR-11-EQPX-0023) and the ERFD European grant SCV-IrDive.”

“This study was supported by the French National Agency for Research (grant ANR-11-EQPX-0023) and the ERFD European grant SCV-IrDive. “

Please note that funding information should not appear in other areas of your manuscript. We will only publish funding information present in the Funding Statement section of the online submission form.

“This study was supported by the French National Agency for Research (grant ANR-11-EQPX-0023) and the ERFD European grant SCV-IrDive.”

4. PLOS requires an ORCID iD for the corresponding author in Editorial Manager on papers submitted after December 6th, 2016. Please ensure that you have an ORCID iD and that it is validated in Editorial Manager. To do this, go to ‘Update my Information’ (in the upper left-hand corner of the main menu), and click on the Fetch/Validate link next to the ORCID field. This will take you to the ORCID site and allow you to create a new iD or authenticate a pre-existing iD in Editorial Manager. Please see the following video for instructions on linking an ORCID iD to your Editorial Manager account: https://www.youtube.com/watch?v=_xcclfuvtxQ.

5. Please upload a new copy of Figure 2 and 3 as the detail is not clear. Please follow the link for more information: " ext-link-type="uri" xlink:type="simple">https://blogs.plos.org/plos/2019/06/looking-good-tips-for-creating-your-plos-figures-graphics/"
https://blogs.plos.org/plos/2019/06/looking-good-tips-for-creating-your-plos-figures-graphics.

Additional Editor Comments (if provided):

The key points across the reviews are:

- Clarifications in introduction, methods and results which are partly rather unclear

- Considering to exclude potentially high prevalence of neutral outputs from the Facereader

- Reviewing the conclusions in the discussion for validity based on the current results

- A thorough proof-reading of your manuscript

Reviewers' comments:

Reviewer's Responses to Questions

**Comments to the Author**

1. Is the manuscript technically sound, and do the data support the conclusions?

Reviewer #1: No

Reviewer #2: Partly

Reviewer #3: Partly

2. Has the statistical analysis been performed appropriately and rigorously? 

Reviewer #1: No

Reviewer #2: Yes

Reviewer #3: N/A

3. Have the authors made all data underlying the findings in their manuscript fully available?

Reviewer #1: Yes

Reviewer #2: Yes

Reviewer #3: Yes

4. Is the manuscript presented in an intelligible fashion and written in standard English?

Reviewer #1: Yes

Reviewer #2: Yes

Reviewer #3: Yes

5. Review Comments to the Author

Reviewer #1: This was a study comparing the facial expressions of people with and without antisocial personality disorder (ASPD) as they recalled important life events. The study adds to just a few others that have examined the production (rather than recognition) of facial expressions in people with ASPD. The Self-Defining Memories task has higher ecological validity than some tasks used by other studies, for example studies that elicited facial expressions by showing participants pictures or videos. I have a number of concerns about the study’s methods – the study’s contribution to the field is currently unclear.

Introduction

Can the authors please define ASPD? What are the features, and how is it distinguished from psychopathy?

Relatedly, the authors should make it clear in their literature review whether prior studies assessed ASPD vs. Psychopathy. For example, the authors cite Künecke et al. (2018) as a study of ASPD, yet the study instead assessed psychopathy. The distinction is important, as behavioral, physiological, and neural correlates of the two diagnostic categories often differ.

The authors note that FaceReader is less obtrusive than electromyography. But doesn’t FaceReader also pick up on contractions unrelated to expressions of emotion and nonverbal communications (p. 4 line 81-83)? Is this also a limitation of FaceReader?

The authors should provide a citation for the theory of six basic emotions (probably a paper by Paul Ekman). However, I’d also recommend discussing more recent data that contradict this theory (for example, Barrett, Adolphs, Martinez, Marsella, Pollak, 2019). These data suggest that six basic emotions are not universally expressed and are not exact readouts of a person’s emotional state.

Did the authors describe all their hypotheses in the introduction? I only see a hypothesis about increased anger expression during SDM retrieval in the ASPD group

Methods

There is not enough information about the control group. How was the control group recruited? Was the control group matched to the ASPD group? Was the control group assessed for ASPD or psychopathy. The control group seems to be younger and more educated than the ASPD group, and these differences do not seem to be controlled for in the analyses.

Why was the SCID-II used to assess ASPD when the current version is the SCID-5?

On p. 10 line 238-239, the authors lay out the rationale for their analytic strategy: “Bayesian analyses corresponded to the best statistical method in view of the sample size and the data of the experimental paradigm” What does this mean? Why, specifically, was this the best method?

Why were mixed effects Bayesian models not used for the PCL-R analyses? Why were PCL-R Total scores not analyzed?

Were corrections for multiple comparisons performed?

Results

I don’t see any description of the types of self-defining memories each group reported. This seems like an important detail. Did the ASPD group report more negative events than the control group? Did they report more intense events than the control group? Did participants provide subjective ratings of their emotional state? My main question: did the group differences arise simply because the ASPD group reported more negative life events (which would likely result in more negative and angry expressions while participants described the events)?

Discussion

It is very difficult to conclude that “the expression of anger in individuals with ASPD cannot be understood only as a simple basic emotion but could correspond to a more stable affective state like mood associated rather with highly saturated emotions” (p. 18 line 371-374), given the lack of information about the control group and the memories that each group recalled. Does the Self-Defining Memories task allow authors to make conclusions about "stable affective states" or just affective states during the task? The authors need to provide more information and control for important differences between the groups before this conclusion can be drawn.

“Neutral state” (p. 19 line 403-404) should be re-written as “neutral expression” to avoid the inference that facial expression corresponds exactly to the participant’s emotional state.

Minor

Please consider splitting up paragraphs based on topic in the introduction.

Since the PCL-R was used, I would recommend using labels for the factors/facets that are typically used for this measure: Facet 1 “Interpersonal,” Facet 2 “Affective,” etc. (see work by Hare).

“Mage” should be “M age” on p. 5 line 115.

"Dichotomous fixe factor" should be changed to “dichotomous fixed factor” on p. 10 line 241.

The authors should consider using person-first or similar language to refer to people in prison or forensic institutions. Instances of “inmate” should preferably be changed to “incarcerated person.” See this helpful resource from the Marshall Project (https://www.themarshallproject.org/2021/04/12/what-words-we-use-and-avoid-when-covering-people-and-incarceration)

Reviewer #2: Thank you for inviting me to review this manuscript which presents a study that examines the facial reactivity during the retrieval of self-defining memories in individuals with Antisocial Personality Disorder (APD) and controls. The study has some interesting findings and used a new methodology to access participants facial reactions (Facereader). There are a number of weaknesses however that affect my enthusiasm for the manuscript that are outlined below in more detail in my review with some suggestions on how to address these.

Reviewer #3: Please see my concerns and open questions regarding the conclusions and implications of the results in the detailed comments. Unfortunately, I am no expert with regard to the statistical analysis that was conducted.

6. PLOS authors have the option to publish the peer review history of their article (what does this mean?). If published, this will include your full peer review and any attached files.

Reviewer #1: No

Reviewer #2: No

Reviewer #3: No

---

## [Author Response · Author response to Decision Letter 0]

13 Dec 2021

You will find all the answers to the reviewers detailed in the response letter.

Best regards

JLouis Nandrino

---

## [Decision Letter · Decision Letter 1]

3 Jan 2022

PONE-D-21-21994R1Monitoring emotional facial reactions of forensic inpatients with antisocial personality disorder during retrieval of self-defining memoriesPLOS ONE

Dear Dr. Nandrino,

Thank you for submitting your manuscript to PLOS ONE. After careful consideration, we feel that it has merit but does not fully meet PLOS ONE’s publication criteria as it currently stands. Therefore, we invite you to submit a revised version of the manuscript that addresses the points raised during the review process.

The revised version of your manuscript has now been reveiwed by the original three reviewers. As you can see below, Reviewer 3 thought their comments and questions were well addressed, but have some remaining questions. Further, there are several open questions raised by Reviewer 1, such as clarifying differentiation between emotional experience and facial muscle activity, or some aspects of the methodology, which you could address in a revision, as well as substantial proof-reading (as suggested by Reviewer 2, an English native speaker might be helpful).

We look forward to receiving your revised manuscript.

Kind regards,

Hedwig Eisenbarth

Academic Editor

PLOS ONE

Journal Requirements:

Additional Editor Comments (if provided):

Thank you for submitting a revised version of your manuscript, which has now been reveiwed by the original three reviewers. As you can see below, reviewer 3 thought their comments and questions were well addressed. However, there are still several open questions raised by Reviewer 1, such as clarifying differentiation between emotional experience and facial muscle activity, or some aspects of the methodology, which you could address in a revision, as well as substantial proof-reading (as suggested by Reviewer 2, an English native speaker might be helpful).

Reviewers' comments:

Reviewer's Responses to Questions

**Comments to the Author**

1. If the authors have adequately addressed your comments raised in a previous round of review and you feel that this manuscript is now acceptable for publication, you may indicate that here to bypass the “Comments to the Author” section, enter your conflict of interest statement in the “Confidential to Editor” section, and submit your "Accept" recommendation.

Reviewer #1: (No Response)

Reviewer #2: (No Response)

Reviewer #3: All comments have been addressed

2. Is the manuscript technically sound, and do the data support the conclusions?

Reviewer #1: No

Reviewer #2: (No Response)

Reviewer #3: Yes

3. Has the statistical analysis been performed appropriately and rigorously? 

Reviewer #1: I Don't Know

Reviewer #2: N/A

Reviewer #3: N/A

4. Have the authors made all data underlying the findings in their manuscript fully available?

Reviewer #1: Yes

Reviewer #2: Yes

Reviewer #3: Yes

5. Is the manuscript presented in an intelligible fashion and written in standard English?

Reviewer #1: Yes

Reviewer #2: No

Reviewer #3: Yes

6. Review Comments to the Author

Reviewer #1: This study adds to just a few others that have examined the production (rather than recognition) of facial expressions in people with antisocial personality disorder (ASPD). The authors have addressed several of the concerns I noted in my previous review. However, some of my concerns remain. I hope the authors can address the points below to ensure that the contribution of their data to our understanding of ASPD is clear.

My main concern is that the authors have concluded from their data that “anger is a core emotion” (lines 459-460) of ASPD. Unfortunately the study design does not afford this broad conclusion. My conclusion from their data is this: when asked to describe important life events, people with ASPD are less likely to describe positive events and more likely to make facial expressions associated with anger.

Introduction

I appreciate that the authors added to their list of hypotheses. However, the last sentence in the paragraph is unclear (lines 114-116). What were the authors’ hypotheses regarding happy, sad, and scared expressions? Also, the authors note in this sentence that they “quantified the part devoted to the non-expression of emotional FE.” How is that related to the authors’ hypotheses?

Methods

The authors have added necessary information about the control group. Now, the authors should show that education level was unrelated to frequency of facial expressions (especially neutral and angry expressions). Since the ASPD and control group differed in education level, and the control group was even recruited from university staff, it is possible that education level is driving the significant differences between groups. In fact, education level is probably not the only socioeconomic factor that differs between ASPD group, which was recruited from a forensic hospital, and the control group, recruited from university staff. Ideally, the authors would note this in the limitations section.

In my previous review I requested that the authors distinguish between ASPD and psychopathy, because they reported findings related to both constructs. Since the authors no longer report analyses of psychopathy in the manuscript, it shouldn’t be necessary to describe the psychopathy measurement in the Methods section.

The authors’ rationale for Bayesian analyses is still unclear to me. What metric did the authors use to determine that Bayesian analyses were the best statistical method? Power analysis? Can the authors point me to a paper showing how to decide between Bayesian analyses and the general linear model considering the sample size and experimental design? I’m afraid the rationale is too vague. My worry is that readers may wonder why the authors chose Bayesian analyses for data that would typically have been analyzed with the general linear model, and that readers may think the authors chose the analyses that supported their hypotheses. Clarifying the rationale for the analyses would help!

Results

In my previous review I asked whether the ASPD group described more negative life events in the SDM task. I thank the authors for clarifying that “Chi-squared tests demonstrated a difference between the two groups but the residual chi-squared test did not show which valence was different between the two samples” (lines 345-347). This means there was a difference between groups, correct? The raw percentages suggest the groups differed in rates of positive stories (ASPD = 22.8%, control = 36.2%) and neutral stories (ASPD = 34.7%, control = 16.2%). I think it would be important to acknowledge these differences, because they may have affected the facial expressions the ASPD group made while describing life events.

FaceReader detected fewer neutral expressions in the ASPD group. Wouldn’t one interpretation of the data be that the ASPD group uses more facial expressions when describing life events? Is that surprising, given the ASPD group were describing more neutral life events? Conversely, it sounds like the more educated control group described more positive life events but made fewer facial expressions when describing those events.

Discussion

Related to my point above, the data do not support this statement: “ASPD participants retrieved as many SDM of each valence as the control sample” (lines 387-388). Please revise, as the groups differed in the rates of positive and neutral stories.

I appreciate that the authors now cite Barrett et al. (2019). However, note that Barrett et al.’s comprehensive review found that facial muscle movement does not reliably express any one emotion category. Based on that finding, what do the current facial muscle data tell us about the subjective feeling states of people with ASPD during this task? Should we assume that “angry expressions” signal an underlying subjective feeling of “anger”? This limitation should be noted. I appreciate that the authors already highlight the need for measuring participants’ subjective feelings during the task.

Minor

Again, I would advise changing “neutral state” (line 223) to “neutral expression” to avoid the inference that facial expression corresponds exactly to a person’s emotional state.

Please revise “consider the estimator to be truly different” (line 313) to “consider the estimator to be most likely different.”

I found this statement unclear: “asking people with APSD about their subjective feelings is particularly complex as they have a deficit in their appreciation of these” (lines 440-441). The phrase “appreciation of subjective feelings” is vague. Please revise.

I would recommend revising this conclusion: “our findings shed new light on the emotional experience” (line 457). See my comments above about what facial muscle movements tell us about a person’s emotional experience.

Reviewer #2: Thank you for inviting me to review a revision on this manuscript which presents a study that examines the facial reactivity during the retrieval of self-defining memories in individuals with Antisocial Personality Disorder (APD) and controls.

Unfortunately there are still a number of weaknesses that affect my enthusiasm for the manuscript with the more pronounced one being the language and syntax errors. Overall the paper would be improved by having an English speaker proofread the paper to improve it and make it easy to read and follow. There are still a lot of errors in the revised manuscript submitted.

a. For example the title should be: “Monitoring the emotional facial reactions of individuals with antisocial personality disorder during the retrieval of self-defining memories”

b. Addressing these errors in the abstract and the main paper would greatly enhance the paper. There are many so I will not address them in the review but these will be picked up by a native English speaker.

Reviewer #3: In my view, the authors have done a good job revising the paper and I think the paper clearly improved. My only remaining concern is related to the clarification in the methods section and my previous comment on the description of the samples. In my opinion, differences in age and education (even if neglectable and not critical for the main results) should be mentioned as a possible limitation. Further, if ASPD or other mental disorders have not been assessed in the control sample, this should be stated and mentioned as a potential limitation.

7. PLOS authors have the option to publish the peer review history of their article (what does this mean?). If published, this will include your full peer review and any attached files.

Reviewer #1: No

Reviewer #2: No

Reviewer #3: No

---

## [Author Response · Author response to Decision Letter 1]

12 Mar 2022

See the attached document entitled "response to reviewers"

---

## [Decision Letter · Decision Letter 2]

26 Apr 2022

PONE-D-21-21994R2Monitoring the emotional facial reactions of individuals with antisocial personality disorder during the retrieval of self-defining memoriesPLOS ONE

Dear Dr. Nandrino,

Thank you for submitting your manuscript to PLOS ONE. After careful consideration, we feel that it has merit but does not fully meet PLOS ONE’s publication criteria as it currently stands. Therefore, we invite you to submit a revised version of the manuscript that addresses the points raised during the review process. Just one minor last edit, as you can see from the suggestions from Reviewer 1, which should be easy to address.We will be happy to accept the manuscript with that small additional change.

We look forward to receiving your revised manuscript.

Kind regards,

Hedwig Eisenbarth

Academic Editor

PLOS ONE

Additional Editor Comments:

Just a minor adjustment as requested by Reviewer 1. We will be happy to accept the manuscript with that small additional change.

Reviewers' comments:

Reviewer's Responses to Questions

**Comments to the Author**

1. If the authors have adequately addressed your comments raised in a previous round of review and you feel that this manuscript is now acceptable for publication, you may indicate that here to bypass the “Comments to the Author” section, enter your conflict of interest statement in the “Confidential to Editor” section, and submit your "Accept" recommendation.

Reviewer #1: (No Response)

2. Is the manuscript technically sound, and do the data support the conclusions?

Reviewer #1: Partly

3. Has the statistical analysis been performed appropriately and rigorously? 

Reviewer #1: Yes

4. Have the authors made all data underlying the findings in their manuscript fully available?

Reviewer #1: Yes

5. Is the manuscript presented in an intelligible fashion and written in standard English?

Reviewer #1: Yes

6. Review Comments to the Author

Reviewer #1: The authors have addressed my previous concerns, although I'm still unsure about one issue regarding the valence of SDMs. The authors performed a chi-squared test that showed that ASPD and control participants differed in the valence of memories (lines 348-349). The follow-up residual chi-squared tests did not reveal which valence differed between the two groups. So the groups differed, but the authors cannot conclude how they differed. Yet the authors state that the data "does not allow us to conclude that there is a significant difference in the production of memories depending on valence" (lines 352-353). That statement seems to inaccurately describe the first chi-squared test. Wouldn't it be fairer to say the data "does not allow us to conclude which valence of SDMs was produced more frequently by either group"? I'm willing to accept that the authors are simply interpreting the data very cautiously.

At the very least, I'd recommend the authors revise the following statement in the discussion (lines 436-437): "it appears that anger notably associated with aversive life events should be the target of psychologic interventions." How can the authors make conclusions about aversive life events if the groups did not differ in terms of SDM valence (above)? Also, the data seem to show that the ASPD group displayed more facial expressions associated with anger while talking about positive memories but not negative or mixed memories (Model 4 in Table 1). I think the sentence in lines 436-437 would be fine if it removed the phrase "associated with aversive life events."

7. PLOS authors have the option to publish the peer review history of their article (what does this mean?). If published, this will include your full peer review and any attached files.

Reviewer #1: No

---

## [Author Response · Author response to Decision Letter 2]

8 May 2022

Minor corrections:

Reviewer #1: The authors have addressed my previous concerns, although I'm still unsure about one issue regarding the valence of SDMs. The authors performed a chi-squared test that showed that ASPD and control participants differed in the valence of memories (lines 348-349). The follow-up residual chi-squared tests did not reveal which valence differed between the two groups. So the groups differed, but the authors cannot conclude how they differed. Yet the authors state that the data "does not allow us to conclude that there is a significant difference in the production of memories depending on valence" (lines 352-353). That statement seems to inaccurately describe the first chi-squared test. Wouldn't it be fairer to say the data "does not allow us to conclude which valence of SDMs was produced more frequently by either group"? I'm willing to accept that the authors are simply interpreting the data very cautiously.

Response: We integrated the sentence proposed by the reviewer : « the data does not allow us to conclude which valence of SDMs was produced more frequently by either group ».

Reviewer #2: At the very least, I'd recommend the authors revise the following statement in the discussion (lines 436-437): "it appears that anger notably associated with aversive life events should be the target of psychologic interventions." How can the authors make conclusions about aversive life events if the groups did not differ in terms of SDM valence (above)? Also, the data seem to show that the ASPD group displayed more facial expressions associated with anger while talking about positive memories but not negative or mixed memories (Model 4 in Table 1). I think the sentence in lines 436-437 would be fine if it removed the phrase "associated with aversive life events."

Response: We decided to delete the part of the sentence: « associated with aversive life events ».

---

## [Editor Report · Decision Letter 3]

10 May 2022

Monitoring the emotional facial reactions of individuals with antisocial personality disorder during the retrieval of self-defining memories

PONE-D-21-21994R3

Dear Dr. Nandrino,

We’re pleased to inform you that your manuscript has been judged scientifically suitable for publication and will be formally accepted for publication once it meets all outstanding technical requirements.

Kind regards,

Hedwig Eisenbarth

Academic Editor

PLOS ONE

Additional Editor Comments (optional):

Congratulations to this nice paper.
---

## [Editor Report · Acceptance letter]

18 May 2022

PONE-D-21-21994R3 

Monitoring the emotional facial reactions of individuals with antisocial personality disorder during the retrieval of self-defining memories 

Dear Dr. Nandrino:

I'm pleased to inform you that your manuscript has been deemed suitable for publication in PLOS ONE. Congratulations! Your manuscript is now with our production department. 

Kind regards, 

on behalf of

Dr. Hedwig Eisenbarth 

Academic Editor

PLOS ONE